# BACKDOOR ATTACK FOR FEDERATED LEARNING WITH FAKE CLIENTS

## ABSTRACT

Federated Learning (FL) is a popular distributed machine learning paradigm that enables collaborative model training without sharing clients' data. Recent studies show that federated learning can be vulnerable to backdoor attacks from malicious clients: such attacks aim to mislead the global model into a targeted misprediction when a specific trigger pattern is presented. Although various federated backdoor attacks are proposed, most of them rely on the malicious client's local data to inject the backdoor trigger into the model. In this paper, we consider a new and more challenging scenario that the attacker can only control the fake clients, who do not possess any real data at all. Such a threat model sets a higher standard for the attacker that the attack must be conducted without relying on any real client data (only knowing the target class label). Meanwhile, the resulting malicious update should not be easily detected by the potential defenses. Specifically, we first simulate the normal client updates via modeling the historical global model trajectory. Then we simultaneously optimize the backdoor trigger and manipulate the model parameters in a data-free manner to achieve our attacking goal. Extensive experiments on multiple benchmark datasets show the effectiveness of the proposed attack in the fake client setting under state-of-the-art defenses.

## 1 INTRODUCTION

In recent years, Federated Learning (FL) (McMahan et al., 2017; Zhao et al., 2018) has become a popular distributed machine learning paradigm, where many clients collaboratively train a global model without sharing their local data. After the clients utilize their local training data to train the local models, FL helps aggregate a shared global model with improved performances. Despite FL's capability of joint model training, its formulation naturally leads to repetitive synchronization between the server and the clients. In particular, since the server has little knowledge or control over the client behaviors, it actually facilitates backdoor attacks (Gu et al., 2019), which aim to mislead the model into a targeted misprediction when a specific trigger pattern is presented by stealthy data poisoning. Backdoor attacks injected by the malicious clients can be easily implemented and hard to detect from the server's perspective.

Backdoor attacks on FL are first studied in (Bagdasaryan et al., 2020; Bhagoji et al., 2019). In these works, malicious clients have strong incentives to compromise the shared global model by uploading the local model trained with poisoned data samples (data attached with triggers pointing to the target class). Prior works discover that the backdoor can survive even when various defenses are deployed (Shejwalkar et al., 2022; Baruch et al., 2019). Some defenses check whether uploaded local model updates or weights contain backdoor-related weights or neurons by anomaly detection (Rieger et al., 2022; Ozdayi et al., 2020). Others are dedicated to purifying the potential backdoor from the global model through approaches such as distillation (Sturluson et al., 2021; Xu et al., 2021) and pruning (Liu et al., 2018).

Recent research on backdoor attacks focuses on improving the stealthiness and effectiveness of attacks to evade the wide variety of federated backdoor defenses. Some works (Xie et al., 2019; Sundar et al., 2022) decompose the trigger pattern into sub-patterns to perform distributed attacks or adaptively resize the trigger during training. One line of research (Fang & Chen, 2023; Lyu et al., 2023) attempts to strengthen the backdoor attack either by optimizing the backdoor-related optimization objective or by directly manipulating the updates obtained from the training. Another

line of research focuses on how backdoor attacks can be implemented under more stringent yet practical settings. Zhang et al. (2022b); Wen et al. (2022) evaluate backdoor attacks in the realistic regime where the number of clients is numerous and not consistently queried by the central server. In terms of the data distribution, Zawad et al. (2021) shows that more non-i.i.d. data distribution may result in overfitting at the local training of benign clients, and thus can be utilized by attackers to disguise themselves and fool skewed-feature based defenses.

Despite the enormous works exploring how to enable backdoor attacks to stealthily bypass various defenses, they share the same critical component, namely the access to local data. With local data, the malicious clients can manipulate the updates (gradients) to achieve their attack goals. However, in the real FL system, compromising the genuine clients with actual training data is usually at a high cost that an attacker cannot afford. Even if the attacker can afford the cost of compromising a few genuine clients, the server may only choose a small proportion of clients for aggregation (i.e., partial participation), rendering it pointless to compromise those genuine clients at a high cost.

As an alternative, attacking with fake clients becomes lucrative for the attackers. A malicious attacker participating in the training does not need to maintain a large amount of genuine clients with possess to real data, but instead may control a group of fake devices (clients). These fake clients perform backdoor attacks locally in a data-free fashion. Since there is no need to actually train a model locally on those fake clients, the attacker can actually create more malicious devices (compared to the traditional federated backdoor attack scenarios) to increase the proportion of fake clients in the client pool, thus having a higher chance of completing the backdoor attack to the global model.

Promising as it sounds, backdoor attacks with fake clients sets a higher standard for the attackers. On one hand, the fake clients' updates should be stealthy enough and look similar to those benign updates to escape potential defense or countermeasures on the server. On the other hand, the malicious attack must also successfully inject the backdoor into the global model to achieve the attacking goal. However, when defense mechanisms are deployed on the server side, it is very hard for the fake clients to disguise their identity due to the lack of training data (cannot produce a similar update to those benign ones). Thus they attacks are more likely to be filtered (by anomaly-detection defenses) or weakened (by robust aggregation defenses).

To solve the above-mentioned challenges, in this paper, we propose **F**ake Client **B**ackdoor **A**ttack (FakeBA), a new backdoor attack scheme that relies on fake clients to circumvent federated backdoor defenses without access to any local training data. Specifically, the fake clients first simulate the activities of benign clients, then make mild modifications on several model parameters to directly inject the backdoor without performing model training or fine-tuning. To rigorously evaluate the robustness of current federated backdoor defenses under FakeBA, we conduct comprehensive experiments and an in-depth case study on the proposed attack against state-of-the-art federated backdoor defenses, we summarize our main contributions as follows:

1. We propose a backdoor attack for federated learning with fake clients, FakeBA, which is the first federated backdoor attack that solely relies on fake clients that do not have access to any local training data. Different from traditional backdoor attacks, our attack simulates benign updates and selectively manipulates several model parameters to directly inject the backdoor.

2. The proposed attack does not possess any real training data, but achieves a higher attack success rate without extra damage to its natural accuracy. The experiment results demonstrate the superiority of the proposed attack over other backdoor attacks without data.

3. We evaluate FakeBA against recent federated backdoor defenses, the result shows that FakeBA can either evade the federated backdoor defenses or lead to a large degradation in natural accuracy when applying the defense. These facts suggest that the threat of backdoor attacks from fake clients is tangible in the practical federated learning system.

## 2 PRELIMINARIES

### 2.1 BACKDOOR ATTACKS FOR FEDERATED LEARNING

In federated learning, the participating clients train their local models and upload them to the server, and the server aggregate these models to get a global model. For $M$ participating clients, each of which has its own dataset $\mathcal{D}_i$ with size $n_i$ and $N = \sum_i n_i$. At the $t$-th federated training round, the

server sends the current global model $\boldsymbol{\theta}_t$ to a randomly-selected subset of $m$ clients. The clients then perform $K$ steps of local training to obtain $\boldsymbol{\theta}_t^{i,K}$ based on the global model $\boldsymbol{\theta}_t$, and send the updates $\boldsymbol{\theta}_t^{i,K} - \boldsymbol{\theta}_t$ back to the server. Now the server can aggregate the updates with some specific rules to get a new global model $\boldsymbol{\theta}_{t+1}$ for the next round. In the standard FedAvg (McMahan et al., 2017) method, the server adopts a sample-weighted aggregation rule to average the $m$ received updates:

$$\boldsymbol{\theta}_{t+1} = \boldsymbol{\theta}_t + \frac{1}{N} \sum_{i=1}^{m} n_i (\boldsymbol{\theta}_t^{i,K} - \boldsymbol{\theta}_t). \tag{2.1}$$

Assume there exists one or several malicious clients with goal to manipulate local updates to inject a backdoor into the global model such that when the trigger pattern appears in the inference stage, the global model would a give preset target prediction $y_{\text{target}}$. In the meantime, the malicious clients do not want to tamper with the model's normal prediction accuracy on clean tasks (to keep stealthy). Therefore, the malicious client has the following objectives:

$$\min_{\boldsymbol{\theta}} \mathcal{L}_{\text{train}}(\mathbf{x}, \mathbf{x}', y_{\text{target}}, \boldsymbol{\theta}) := \frac{1}{n_i} \sum_{k=1}^{n_i} \ell(f_{\boldsymbol{\theta}}(\mathbf{x}_k), y_k) + \lambda \cdot \ell(f_{\boldsymbol{\theta}}(\mathbf{x}_k'), y_{\text{target}}), \tag{2.2}$$

where $\boldsymbol{\Delta}$ denotes the associated trigger pattern, and $\mathbf{x}_k' = (\mathbf{1} - \mathbf{m}) \odot \mathbf{x}_k + \mathbf{m} \odot \boldsymbol{\Delta}$ is the data attached with trigger for the backdoor task, $\mathbf{m}$ denotes the trigger location mask, and $\odot$ denotes the element-wise product. The first term in equation 2.2 is the common empirical risk minimization while the second term aims at injecting the backdoor trigger into the model. The loss function $\ell$ is usually set as the CrossEntropy Loss. $\lambda$ controls the trade-off between the two tasks.

## 2.2 DIFFICULTY OF BACKDOOR ATTACKS WITH FAKE CLIENTS

Most existing backdoor attacks on FL (Bagdasaryan et al., 2020; Bhagoji et al., 2019; Xie et al., 2019) are based on training over triggered data samples equation 2.2. This requires the malicious client to have (part of) training data $\mathbf{x}$ locally so that they can attach triggers $\boldsymbol{\Delta}$ to implant a backdoor. However, for fake clients without any training data locally, such an attack mode is completely inapplicable: First, the malicious clients cannot attach the trigger without training samples, and hence cannot perform local training or bind the trigger to a target label. Secondly, when the server applies various backdoor defenses, with no information on the training data distribution, it becomes harder to make their updates stealthy. If fake clients need to perform a backdoor attack, they need to carefully design a method to achieve stealthiness and keep attack utility simultaneously.

To the best of our knowledge, there is no research that has yet found an attack strategy with fake clients. Some naive strategies may resort to simulating a dataset via model inversion attack (MI) (Fredrikson et al., 2015) or introducing substitution dataset (SD) (Lv et al., 2023). Model inversion attempts to recreate data samples through a predictive model by minimizing the loss of data on the preset labels, while the substitution dataset directly finds another dataset similar to the original dataset[1]. Although the above two strategies can provide the fake clients with simulated local datasets, there are still several major downsides: (1) model inversion cannot precisely reconstruct batches of input samples without the confidence value of prediction, also the inversed dataset is still quite different from the real one and hence cannot directly replace the real data in the case of model training; (2) As the distribution of the introduced dataset and the real training dataset are not exactly the same, crafting a backdoored local model with substitution dataset usually requires a huge amount of instances to avoid performance degradation caused by over-fitting or catastrophic forgetting, and it is also costly and time-consuming for fake clients to apply dataset reduction skills (Lv et al., 2023).

## 3 PROPOSED METHOD

In this section, we describe our proposed FakeBA attack. First, we introduce the threat model considered in our paper.

---

[1]For example, when each of the benign clients holds a non-overlapping portion of the ImageNet data as local training samples, the fake clients can introduce CIFAR-100 as a substitution dataset to perform backdoor injection as shown in equation 2.2.

## 3.1 THREAT MODEL

**Attacker's Goal:** The attacker's goal is to inject a backdoor trigger to the global model: The global model can still have a high prediction accuracy on the clean samples, when facing a data sample with the specific trigger, it will give a prediction to the target class. Generally speaking, the attacker wants to keep a high attack success rate on triggered data samples as well as keep a high test accuracy on the clean samples.

**Attacker's Capability:** We assume the attacker can inject a certain amount of fake clients into FL systems, and can control the fake clients to send arbitrary fake updates to the server. when necessary, these fake clients can pass information to each other (collude). Performing the attacks with these fake devices requires much less computing resources as these devices do not really need to train on local data (the server lacks the knowledge on these fake clients' activities during local training).

**Attacker's Knowledge:** We suppose that the injected fake clients do not have training samples locally. Instead, the fake clients can simulate the activities of benign clients to disguise themselves. The scenario is practical since the server can only get the trained model from clients without the information on how the model is trained (or even crafted) and whether the received local models are genuinely trained on actual data samples. Correspondingly, the malicious attacker is unable to influence the operations conducted on the central server such as changing the aggregation rules (defense schemes), or tampering with the model updates of other benign clients.

## 3.2 FAKE CLIENT BACKDOOR ATTACK FOR FEDERATED LEARNING

As is discussed in Section 2.2, constructing fake clients entirely without data poses three main challenges: 1) Fake clients need to perform like benign clients to bypass potential defenses/countermeasures; 2) The inserted backdoors need to achieve the attack objective without any real training data; 3) For the scenario of FL with partial-participating clients, ensure that the fake clients' attack strategy is still applicable.

We carefully design our proposed FakeBA method tailored to these three challenges. First, we simulate benign updates through the trajectory of historical global models over multiple rounds. Then, we insert backdoors by meticulously analyzing and modifying model parameters, thereby embedding backdoors into the simulated benign parameters. Lastly, we extend our method to accommodate the scenario of FL with partial-participating clients. Following this, we will discuss in detail how we address these challenges.

Suppose the $i$-th client is a fake client, let us denote its received global model at the $t$-th round as $\boldsymbol{\theta}_t := \{\mathbf{w}^{[1]}, \mathbf{w}^{[2]}, .., \mathbf{w}^{[L]}\}$ and each layer's output as $\mathbf{z}^{[1]}(\cdot), \mathbf{z}^{[2]}(\cdot), .., \mathbf{z}^{[L]}(\cdot)$. We take the full-participation circumstance (all clients participate in the aggregation for each round) as an example and introduce the details of FakeBA.

**Simulating the Benign Updates on Fake Clients** Our first step is to simulate the benign updates on fake clients to disguise their identities. This is necessary since many defenses (Cao et al., 2021; Zhao et al., 2022; Rieger et al., 2022) are adopting similarity-based anomaly detection techniques to check whether the local updates are significantly different from the majority (other clients' updates). In order to simulate the benign updates, FakeBA aims to utilize the fake clients' received historical global models. Denote the model update difference as $\Delta\boldsymbol{\theta}_t = \boldsymbol{\theta}_t - \boldsymbol{\theta}_{t-1}$. Based on the Cauchy mean value theorem (Lang, 1964), we have the approximation of the global update in the $t$-th round:

$$\boldsymbol{g}_t = \boldsymbol{g}_{t-1} + \widehat{\mathbf{H}}_t \cdot (\boldsymbol{\theta}_t - \boldsymbol{\theta}_{t-1}), \qquad (3.1)$$

where $\boldsymbol{g}_t$ is the gradient of global model, $\widehat{\mathbf{H}}_t$ is an integrated Hessian for global model updates in the $t$-th iteration. In practice, the participating clients usually adopt stochastic gradient descent (SGD) as their local optimizer and perform multiple steps of SGD update locally before sending back the updates to the server. Yet we can simply approximate the gradient here using the model difference $\boldsymbol{g}_t \approx \boldsymbol{\theta}_t - \boldsymbol{\theta}_{t-1}$. Meanwhile, we denote the global model update difference as $\nabla\boldsymbol{g}_t = \boldsymbol{g}_t - \boldsymbol{g}_{t-1}$. If observing the pairs of $\nabla\boldsymbol{g}_t$ and $\boldsymbol{g}_t$, we can actually approximate the value of $\widehat{\mathbf{H}}_t$. Specifically, let the latest $Q$ rounds of model differences and model update differences from the $t$-th round $\nabla\boldsymbol{\Theta}_t = \{\nabla\boldsymbol{\theta}_{t-Q}, \nabla\boldsymbol{\theta}_{t-Q+1}, \ldots, \nabla\boldsymbol{\theta}_{t-1}\}$ and $\nabla\mathbf{G}_t = \{\nabla\boldsymbol{g}_{t-Q}, \nabla\boldsymbol{g}_{t-Q+1}, \ldots, \nabla\boldsymbol{g}_{t-1}\}$. The fake clients can adopt the L-BFGS algorithm to estimate $\widehat{\mathbf{H}}_t$: $\widehat{\mathbf{H}}_t \approx$ L-BFGS$(\nabla\boldsymbol{\Theta}_t, \nabla\mathbf{G}_t)$. With

the approximated $\widehat{\mathbf{H}}_t$, the fake client can simulate the global update $\widehat{\boldsymbol{g}}_t$ as:

$$\widehat{\boldsymbol{g}}_t = \boldsymbol{g}_{t-1} + \widehat{\mathbf{H}}_t \cdot (\boldsymbol{\theta}_t - \boldsymbol{\theta}_{t-1}). \tag{3.2}$$

When the L-BFGS algorithm estimates the integrated Hessian precisely, the simulated benign update is expected to be close to the global model update, thus our fake clients can simulate the benign update the disguise their identities in this way.

**Backdoor Injection** Our next goal is to inject the backdoor into the model parameters in a data-free fashion. Previous studies usually injected backdoors by training the model on triggered data. For fake clients that do not access data, we need to explore other ways to inject the backdoors. Our intuition here is to directly modify a small number of model parameters such that the modified model would have a strong activation pattern (towards the target class) when facing any data with the trigger pattern.

To achieve this, FakeBA starts by optimizing the backdoor trigger such that one specific neuron in the first layer of the model would be very sensitive to the presence of the trigger, compared with other neurons. For an initialized trigger pattern $\boldsymbol{\Delta}$, suppose we are to maximize its activation on the first neuron $\mathbf{z}_1^{[1]}$ (i.e., maximizing $\sum_p (w_p \Delta_p + b)$ where $b$ denotes the bias for the neuron and each $w_p$ denotes the model parameters connected to the neuron), we have:

$$\Delta_p = \begin{cases} \alpha_l, & \text{if } w_p \leq 0 \\ \alpha_u, & \text{otherwise} \end{cases}, \tag{3.3}$$

where $\alpha_l$ and $\alpha_u$ are the upper and lower bound of element values for input samples $\mathbf{x}$. Given fixed model parameters, the optimized backdoor trigger $\boldsymbol{\Delta}$ will have the largest activation for that specific neuron in the first layer.

Next, we aim to modify several parameters in the following layers to ensure the model output target class when facing the optimized trigger and finish the attack. Consider the first neuron's output, i.e., $\sigma(\sum_p (w_p \Delta_p + b))$, which is maximized for trigger $\boldsymbol{\Delta}$. Our goal here is to amplify its activation layer by layer until the model outputs $y_{\text{target}}$ when trigger $\boldsymbol{\Delta}$ is presented. Again, we first select certain neurons along which the activation for the trigger would be enlarged. Suppose $s^{[l]}$ is the selected neuron in the $l$-th layer, where $l = 2, 3, ..., L-1$. We add $\gamma \|\widehat{\boldsymbol{g}}_t\|_\infty$ on the parameter weight between $s^{[l]}$ and $s^{[l+1]}$ [2] where $\gamma$ is a hyperparameter controlling the strength of the attack. As the global model converges, $\|\widehat{\boldsymbol{g}}_t\|_\infty$ becomes smaller, and thus our parameter manipulation (i.e., $\gamma \|\widehat{\boldsymbol{g}}_t\|_\infty$) also becomes less obvious. This design naturally restricts the strength of our parameter manipulation as the training progresses and keeps our attack stealthy.

**Extension to Partial Participation** FakeBA can similarly be extended to FL with partial-participating clients where only part of the clients will be selected for aggregation in each training round. The main difference with the full-participation FL is the definition of the historical model differences $\nabla \boldsymbol{\Theta}_t$ and historical model update differences $\nabla \mathbf{G}_t$ used in L-BFGS algorithm: In some rounds for partial-participation FL, all selected clients are benign, and the fake clients can not get the global model weights. In other cases, we have at least one malicious fake client being selected for participation. Under such cases, since all the fake clients are controlled by a joint attacker and thus can share their received global models and form a shared list of $\nabla \boldsymbol{\Theta}_t$ and $\nabla \mathbf{G}_t$. We can similarly apply the L-BFGS algorithm to simulate benign updates and then attempt to backdoor the global model following the same way as in full participation scenario.

## 4 EVALUATING FAKE CLIENT BACKDOOR ATTACK ON STATE-OF-THE-ART FEDERATED BACKDOOR DEFENSES

### 4.1 DATASETS AND EXPERIMENT SETUPS

We evaluate our FakeBA on several state-of-the-art backdoor defenses for federated learning. Based on the defense mechanism, we classify the tested federated backdoor defenses into two major cat-

---

[2]The manipulation can similarly apply to convolutional layers: if the $s^{[l]}$-th filter is compromised in the $l$-th layer, the compromised parameter in the $l+1$-th layer is in the central of the $s^{[l+1]}$-th filter' $s^{[l]}$-th channel

egories: *Byzantine-robust federated backdoor defenses* [3] and *Non-byzantine-robust federated backdoor defenses*. We test our attack on CIFAR-10 dataset (Krizhevsky & Hinton, 2009) and Tiny-ImageNet dataset (Le & Yang, 2015) under the non.i.i.d. data distribution. The performances of the federated backdoor attacks are measured by two metrics: Attack Success Rate (ASR), i.e., the proportion of triggered samples classified as target labels and Natural Accuracy (ACC), i.e., prediction accuracy on natural clean examples. We test the global model after each round of aggregation: use the clean test dataset to evaluate ACC, average all the optimized triggers as a global trigger and attach it to the test dataset for ASR evaluation.

## 4.2 ATTACK SETTINGS

Our goal is to evaluate the probability of backdooring a global model in FL with fake clients. We compare the FakeBA with Model Inversion (MI) and Substitution Dataset (SD) to test the effectiveness. In each training round, we randomly pick half of the clients from a total of 40 clients (4 are fake clients) to participate. The chosen benign clients train two epochs for their local data. The fake clients can collude before they inject the backdoors into their local models. For more details, we set the non-i.i.d. data with the concentration parameter $h = 1.0$ for our training dataset.

To show the impact of FakeBA on the model performance, in Table 1, we list the clean ACC of the global model under these defenses before and after the FakeBA attack, which shows that FakeBA causes only little loss of ACC on most defenses (0.5% to 1%).

| CIFAR-10 | FedDF | FedRAD | Fine-tuning | RobustLR | DeepSight | FedInv | FLTrust | Bulyan |
|----------|-------|--------|-------------|----------|-----------|--------|---------|--------|
| **Before** | 73.65 | 72.81 | 70.96 | 68.55 | 73.05 | 72.58 | 70.02 | 65.92 |
| **After** | 73.14 | 72.20 | 70.90 | 68.17 | 72.91 | 72.50 | 69.81 | 66.10 |
| **Tiny-Imagenet** | **FedDF** | **FedRAD** | **Fine-tuning** | **RobustLR** | **DeepSight** | **FedInv** | **FLTrust** | **Bulyan** |
| **Before** | 32.45 | 31.88 | 29.92 | 30.90 | 32.01 | 32.02 | 30.08 | 27.90 |
| **After** | 31.80 | 31.65 | 29.79 | 30.59 | 31.64 | 32.01 | 28.84 | 27.25 |

Table 1: The clean accuracy before and after FakeBA attack.

## 4.3 ATTACKING NON-BYZANTINE-ROBUST FEDERATED BACKDOOR DEFENSES

**FakeBA can evade distillation:** Distillation leverages unlabeled data to achieve robust server-side model fusion, aggregate knowledge from the received clients' models. Figure.1(a) and figure.1(b) show the result of FakeBA against FedRAD (Reddi et al., 2020) and FedDF (Lin et al., 2020). The results show that FakeBA achieves high ASRs on two distillation-based defenses. FedDF and FedRAD are designed to overcome data drift, yet their enhanced model robustness cannot fully purify the backdoor injected. Since we maximize the activation on the neurons layer by layer, the incurred high activation value towards the target label cannot easily erased (especially when the server do not know the actual trigger pattern). Also, the server does not filter fake models before aggregation, causing the predictions of these models for unlabeled samples to be aggregated as well, which may hinder the purification.

**FakeBA can evade fine-tuning:** We also experiment to purify the backdoor through directly fine-tuning the global model with labeled data. Despite with the ground-truth labels, we find that fine-tuning still fails to defend our attack; Without knowing the actual trigger trigger, the client is unable to target the compromised backdoor-associated model parameters and their activated neurons (same as distillation-based defenses); Due to the nature of the non-i.i.d. data distribution in federated learning, the server's labeled data may be significantly deviate from those on the clients' sides. Based on the results defending our attack with fine-tuning leads to more the natural accuracy loss(5% ACC damage compared with defending with distillation).

**FakeBA keeps high natural accuracy:** Based on figure.1, FakeBA improves more ASRs than Substitution Dataset and Model Inversion, but does not invoke an additional loss of ACC. Specifically, when attack against fine-tuning, the global model injected with backdoor FakeBA retains apparently

---

[3]Byzantine-robust federated backdoor defenses come from the utilization of Byzantine-robust aggregation methods. Some of the Byzantine-robust aggregation methods are originally proposed for defending model poisoning attack (Byzantine robustness) yet they may also be used for defending backdoors.

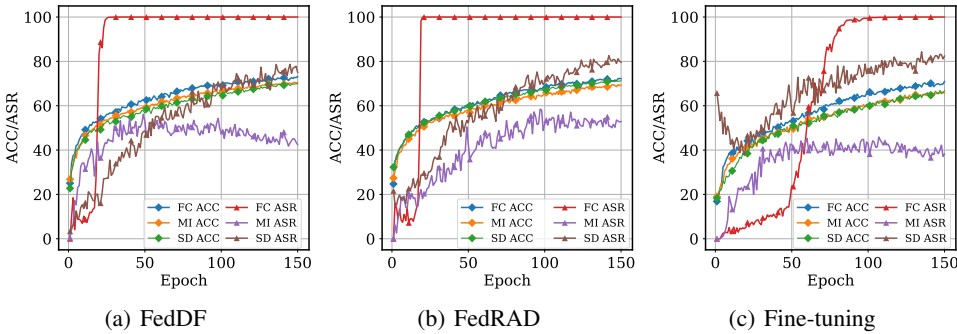

Figure 1: FakeBA (FC) against Non-Byzantine-robust federated backdoor defenses

higher ACC. This may be due to the fact that FakeBA's simulated fake updates follow the trajectory of historical global model differences, which better overcomes the data drift brought by the server-side fine-tuning.

**FakeBA outperforms Model Inversion and Substitution Dataset:** We compare FakeBA with Model Inversion and Substitution Dataset. Figure.2 shows that, when backdoor the global model against the Non-byzantine-robust Federated Backdoor Defenses, our FakeBA incurs small classification loss and achieves much higher ASR (nearly 100%) than other methods. Since fake clients do not need to perform local trianing, their backdoor injection is much less time-consuming. In general, FakeBA can backdoor the current Non-byzantine-robust federated backdoor defenses more efficiently.

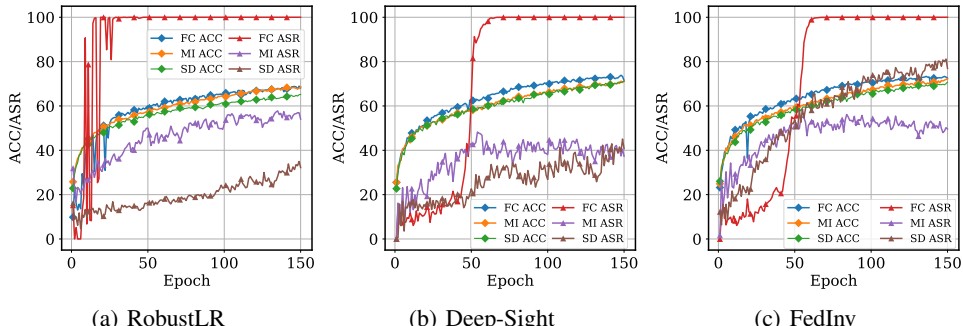

Figure 2: FakeBA (FC) against Similarity-based Byzantine-robust federated backdoor defenses

### 4.4 ATTACKING BYZANTINE-ROBUST FEDERATED BACKDOOR DEFENSES

**FakeBA can bypass similarity-based aggregation rules:** A portion of the Byzantine-robust federated backdoor defenses use similarity metrics to exclude anomalous updates uploaded by fake clients during aggregation (Zhao et al., 2022; Cao et al., 2021; Rieger et al., 2022). The server first computes the pairwise distance (similarity) for the participating clients, ant then exclude the updates far deviating from others. Figure 2 shows that FakeBA is quite resilient to similarity-based aggregation rules. Since it manipulates only a few model parameters, the uploaded fake updates would not be significantly different from the benign updates under the criteria of cosine similarity (Deep-Sight, FLTrust) and Wasserstein distance (FedInv). Consequently, to reduce the ASR of the FakeBA, the server needs to enforce stronger robust aggregation rules.

**FakeBA can evade Robust Learning Rate (RobustLR):** RobustLR(Ozdayi et al., 2020) works by adjusting the servers' learning rate based on the signs of clients' updates: it requires a sufficient number of votes on the signs of the updates for each dimension to move towards a particular direction. Figure.2(a) demonstrates that our attack is effective against such a sign-flipping defense mechanism. It is easy to understand since FakeBA only compromises a very small fraction of model parameters, it would not largely change the voting outcome. As long as the updates on the com-

promised parameter may not be reversed in most rounds, the weght on the position of compromised parameters still gradually accumulate, and let the global model finally be backdoored.

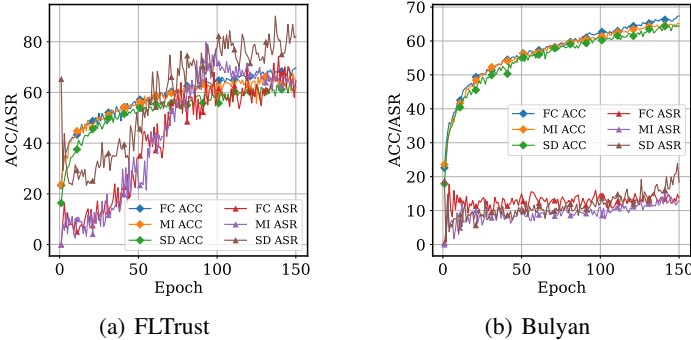

| (a) FLTrust | (b) Bulyan |

Figure 3: FakeBA against Byzantine-robust federated backdoor defenses

**Defending FakeBA sacrifices more natural accuracy:** We test FakeBA on FLTrust and Bulyan. FLTrust excludes the updates with negative cosine-similarity to its benchmark model update trained from the server-side labeled data. For updates not excluded, it shrinks them to the same magnitude as the benchmark global model update. Bulyan excludes clients' updates far from the median of all the proposed updates coordinate-wisely. Figure.3(a) and figure.3(b) show that the two defenses are resilient to FakeBA. Bulyan can completely defend FakeBA due to the fact that FakeBA's updates on the compromised parameters are usually larger, thus are likely to be targeted by the coordinate-based exclusion. Shrinking the updates as FLTrust also prevents the fake clients from dominating the aggregated updates. However, these two defenses hinder federated learning to fully exploit the data heterogeneity. Our experiments find that they lead to lower natural accuracy.

## 4.5 ABLATION STUDY

**Impact of L-BFGS:** We test the effectiveness of L-BFGS-simulated updates in the perspective of cosine-similarity, since cosine-similarity has been widely applied to measure and exclude anomaly updates (i.e., in FLTrust and Deep-Sight). For each client's update, we compute the sum of the cosine similarities with other updates in figure.4. We switch the defense from Deep-Sight to plain FedAvg to avoid the clients being excluded. We compare the simulated updates estimated by LBFGS using different number of historcial global models. The result shows that L-BFGS-simulated updates are relatively more closed to other updates than a single benign client's updates, and also better than using the last model difference (using one model).

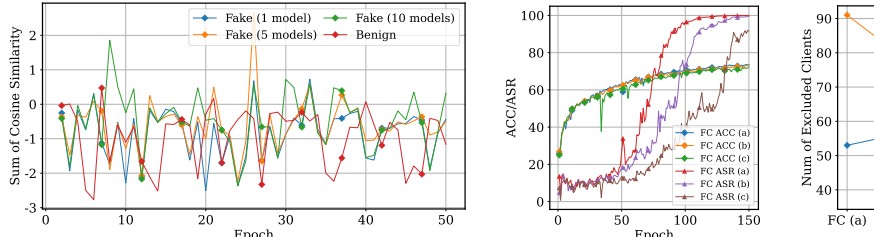

Figure 4: Cosine similarity between the updates.

Figure 5: FakeBA against Deep-Sight under collusion

Figure 6: Number of excluded updates

**Impact of Collusion between Fake Clients:** We test FakeBA against Deep-Sight under three cases: a) one attacker controls 4 fake clients; b) two attackers, each controls two disjointed fake clients); c) four attackers, each controls a fake clients. An attacker's fake clients can collude during the federated learning. Figure.5 and Figure.6 show that a greater degree of collusion leads to higher ACCs and ASRs, which bodes well for the feasibility of attackers implanting more fake clients. As for attack against Deep-Sight, wider collusion can increase the number of excluded benign clients yet decrease the number of excluded fake clients. If there are more fake clients to collude with, they

are more likely receive the global model at each round (be selected for aggregation). As a result, the historical global model information inputted to L-BFGS are in the consecutive rounds, which help the simulated benign updates become more closed to real benign updates.

## 5 ADDITIONAL RELATED WORK

In this section, we review the most relevant works in general FL as well as the backdoor attack and backdoor defenses of FL.

**Federated Learning**: Federated Learning (Konečnỳ et al., 2016) was proposed for the privacy-preserving machine learning in distributed settings. FedAvg (McMahan et al., 2017) works by averaging local SGD updates, of which the variants have also been proposed such as SCAFFOLD (Karimireddy et al., 2020), FedProx (Li et al., 2020), FedNova (Wang et al., 2020b). Reddi et al. (2020); Wang et al. (2022) proposed adaptive federated optimization methods for better adaptivity. Recently, new aggregation strategies such as neuron alignment (Singh & Jaggi, 2020) or ensemble distillation (Lin et al., 2020) have also been proposed.

**Backdoor Attacks on Federated Learning**: Bagdasaryan et al. (2020) injects the backdoor by predicting the global model updates and replaces them with the one that was embedded with backdoors. Bhagoji et al. (2019) aims to achieve global model convergence and targeted poisoning attacks by explicitly boosting the malicious updates and alternatively minimizing backdoor objectives and the stealth metric. Wang et al. (2020a) shows that robustness to backdoors implies model robustness to adversarial examples and proposed edge-case backdoors. DBA (Xie et al., 2019) decomposes the trigger into sub-patterns and distributes them for several malicious clients to implant. Different from traditional training (on triggered data) and rescaling (the malicious client model) based backdoor injection, F3BA (Fang & Chen, 2023) directly modifies (a small proportion of) local model weights to inject the backdoor trigger via sign flips, and jointly optimizes the trigger pattern with the client model. Besides the effectiveness, Neurotoxin (Zhang et al., 2022b) selectively optimizes parameters with maximum update magnitude to boost the durability of current backdoor attacks.

**Backdoor Defenses on Federated Learning**: Robust Learning Rate (Ozdayi et al., 2020) flips the signs of some dimensions of global updates. Wu et al. (2020) designs a collaborative pruning method to remove redundant neurons for the backdoor. Xie et al. (2021) proposed a certified defense that exploits clipping and smoothing for better model smoothness. BAFFLE Andreina et al. (2021) uses a set of validating clients, refreshed in each training round, to determine whether the global updates have been subject to a backdoor injection. Recent work (Rieger et al., 2022; Zhang et al., 2022a) identifies suspicious model updates via clustering-based similarity estimations or integrated Hessian. Another related line of research is Byzantine Robust FL which may also be helpful here (Yin et al., 2021; Fung et al., 2018; Cao et al., 2021). Krum (Blanchard et al., 2017) chooses local updates most similar to the global update, while Trimmed Mean (Yin et al., 2018) aggregates the model parameters by coordinates after discarding the ones with maximum or minimum values. Bulyan (Guerraoui et al., 2018) integrates both Krum and Trimmed Mean to iteratively exclude updates. Pillutla et al. (2019) proposed to replace the weighted arithmetic mean with a geometric median. Besides designing aggregation rules, gradient inversion is also used to reverse the clients' local data, of which the pairwise similarity becomes the metric of aggregation rules (Zhao et al., 2022).

## 6 CONCLUSIONS

In this work, for the first time, we develop a backdoor attack with fake clients. Different from traditional federated backdoor attacks, our fake client attack do not possess any training data. Specifically, the fake clients first simulate the benign updates based on the historical global model updates and then selectively compromise a small proportion of parameters to inject the backdoor trigger in a data-free fashion. The experiment shows that our FakeBA attack can successfully evade multiple state-of-the-art defenses under mild assumptions, while the methods that can defend against it come at a high cost in terms of natural accuracy. Moreover, we perform comprehensive ablation studies and find that the critical design in our method significantly boosts the attack. An interesting future work is designing attacks that can actually defeat the median-based Byzantine robust backdoor defenses, and the flip side is to look for defenses that can defend FakeBA with less damage on the natural accuracy.

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
