# OpenReview forum: "Backdoor Attack for Federated Learning with Fake Clients"
_ICLR.cc/2024/Conference — ICLR 2024 Conference Withdrawn Submission_

### Official Review · Reviewer_DwZd · 2023-10-24

**Soundness:** 3 good
**Presentation:** 3 good
**Contribution:** 3 good
**Rating:** 5
**Confidence:** 4

**Summary:**

In this paper, the authors propose a novel backdoor attack using fake clients for federated learning. The fake clients can inject the backdoors into the global model without possessing real local data and achieve the attacking goal while evading most of the existing defenses. Specifically, to ensure the stealthiness of malicious updates and evade detection, they simulate the normal client updates via modeling the historical global model trajectory. Then, for backdoor injection, with the estimated normal updates as a regularization, they optimize the backdoor trigger and manipulate the selected model parameters simultaneously to achieve the attacking goal. Extensive experiments are conducted to demonstrate their method can achieve high attack success rate while being resilient to most defenses.

**Strengths:**

	This paper proposes a backdoor attack for federated learning with fake clients, which can achieve the attack without the need for real data while evading most of the existing defenses.

	Comprehensive experiments are conducted, and the results seem solid.

	The proposed attack is more realistic and more challenging than traditional backdoor attacks for federated learning.

**Weaknesses:**

	There are many noticeable errors in the paper, including incorrect figure references (Sec. 4.3 FakeBA outperforms Model Inversion and Substitution Dataset Figure.2 -> Figure.1), and typos (a lot). Some notation is arranged in an incorrect order (the “Delta” below Eq.2.2)

	Insufficient details or code for reproducibility.

	The estimation for the Hessian matrix using L-BFGS may require heavy computation overhead when model parameters are huge, and the number Q is large.

**Questions:**

	In Sec 3.2, Simulating the Benign Updates on Fake Clients, Since the gradients are simply approximated using the global model difference, am I correct that you only store the received global model parameters to calculate model differences and model update differences to approximate Hessian using L-BFGS? And how many rounds Q are used in the experiments section? Is it possible to overcome the computation overhead for models with huge amounts of parameters?

	In Sec 3.2, Backdoor Injection How do you select neurons to manipulate? Randomly or there are some further details? How many of them in each layer are selected? You may provide more details about it.

	In the experiments section, the results of all figures are based on which datasets? You may include this detail.

	In the ablation study, to study the impact of L-BFGS, why don’t you compare the accuracy, ASR under different defenses using different number of historical updates?

---

### Official Review · Reviewer_9itP · 2023-10-31

**Soundness:** 2 fair
**Presentation:** 2 fair
**Contribution:** 2 fair
**Rating:** 3
**Confidence:** 3

**Summary:**

The authors suggest a method for conducting a backdoor attack in federated learning, using fake clients that lack access to local training data. This approach can elevate the presence of malicious devices within the system. Experimental findings demonstrate the covert nature and success of this suggested attack.

**Strengths:**

1. The idea of directly altering activation patterns of neurons to target a specific class is interesting.

2. The proposed method demands less information compared to traditional backdoor attacks in the context of federated learning.

**Weaknesses:**

1. I find it challenging to accept the assertion that attackers can compromise more devices when they lack access to local training data. In practice, the proportion of malicious devices appears to be unrelated to the computational power of the attackers.

2. The proposed method involves solving an optimization problem to maximize the activation of the first neuron during each federated learning epoch. I recommend that the authors not only assess the effectiveness but also the efficiency of their backdoor injection method, comparing it to local training-based backdoor attacks using poisoned data. Additionally, considering pruning-based backdoor defense mechanisms like [1] and [2], which can eliminate suspected neurons, it would be interesting to evaluate how the proposed attack performs in the presence of such defenses.

3. It's worth noting that historical model updates received from the server can vary significantly from updates of benign clients, especially when aggregation-based defenses are applied or in datasets with a high degree of non-i.i.d. distribution.

4. To provide a comprehensive assessment, I suggest the authors establish baselines by comparing their proposed attack with state-of-the-art backdoor attacks (those mentioned in the related work) that utilize local training. This would help illustrate whether the proposed attack can achieve a similar level of attack effectiveness.


[1] Liu, K., Dolan-Gavitt, B. and Garg, S., 2018, September. Fine-pruning: Defending against backdooring attacks on deep neural networks. In International symposium on research in attacks, intrusions, and defenses (pp. 273-294). Cham: Springer International Publishing.
[2] Wu, C., Yang, X., Zhu, S. and Mitra, P., 2020. Mitigating backdoor attacks in federated learning. arXiv preprint arXiv:2011.01767.

**Questions:**

1. I appreciate the underlying idea behind the proposed backdoor injection optimization. However, it would enhance the clarity of the approach to establish a connection, either through theoretical or empirical means, between this optimization and the original backdoor objective outlined in equation (2.2).

2. It is essential to provide more comprehensive details about the baseline methods, specifically the model inversion attack (MI) and the substitution dataset (SD). Explaining the rationale for choosing these particular methods for comparison would further elucidate the comparative analysis.

3. To provide a more comprehensive evaluation, additional ablation studies are warranted. These could encompass exploring the efficiency of the proposed attack, varying the number of fake clients, and assessing its performance under different levels of non-i.i.d. data distribution.

---

### Official Review · Reviewer_JM2E · 2023-10-31

**Soundness:** 2 fair
**Presentation:** 2 fair
**Contribution:** 2 fair
**Rating:** 3
**Confidence:** 4

**Summary:**

The paper considers a new FL-backdoor attack scenario where the attackers do not possess any real data. To achieve the backdoor attack goal with fake clients, the authors propose the attack FakeBA that simulates benign updates and selectively manipulates several model parameters to directly inject the backdoor. FakeBA simulates the benign updates based on the Cauchy mean value theorem and L-BFGS algorithm, and injects the backdoor trigger by modifying the parameter of a selected neuron in each network layer. The paper conducts experiments to demonstrate that the proposed attack can evade multiple defenses, and also highlight the impact of L-BFGS and collusion between fake clients by ablation studies.

**Strengths:**

* The paper considers the backdoor attack under federated learning, which is an important research area. The scenario where the attacker does not possess training samples is new.
* The paper conducts experiments and ablation studies to demonstrate the performance of the proposed attack.

**Weaknesses:**

* The paper only considers the case where the attack only tends to inject backdoors to one target class, which may not be the case in practice. The proposed methods for injecting backdoors cannot directly extend to multiple target label scenarios.
* The paper does not analyze the performance of L-BFGS algorithm, which significantly influences the success of the attack.
* The experiments are conducted with a limited set of only 40 clients, which may not fully represent real-world scenarios where federated learning typically involves thousands of clients.

**Questions:**

* On page 4, why 'we can simply approximate the gradient here using the model difference $g_t≈ \theta_t− \theta_{t−1}$'? If so, why use L-BFGS to approximate $g_t$?
* Is $\nabla G$ determined by $\nabla \Theta$?
* How does L-BFGS work? The paper does not describe the algorithm.
* During the backdoor injection, why should the parameters in all layers be modified?
* After modifying a neuron in the first layer, will the other neurons also be sensitive to the trigger?

---

### Official Review · Reviewer_Sevc · 2023-11-01

**Soundness:** 2 fair
**Presentation:** 3 good
**Contribution:** 3 good
**Rating:** 5
**Confidence:** 4

**Summary:**

The paper introduces a method that simulates normal client updates while concurrently optimizing the backdoor trigger and manipulating model parameters in a data-free manner. Extensive experiments on various benchmark datasets highlight the effectiveness of this attack under cutting-edge defenses, shedding light on the evolving landscape of adversarial techniques in Federated Learning.

**Strengths:**

The attack evades many state-of-the-art defenses and outperforms other data-free attacks. The ablation studies provide useful design insights. The evaluations on multiple datasets and defenses are quite comprehensive.

**Weaknesses:**

- It is unclear how the backdoor triggers are injected via sending fake client parameters to the server.
- How likely the attacker is to pass the anomaly detection. For example, how many rounds after anomaly detection is launched the sever found the fake client?
- This threat model demands more sophistication from the attacker, necessitating an attack conducted without relying on genuine client data and ensuring the resulting malicious update remains undetectable by potential defenses.

**Questions:**

- In what real-world scenario can this attack be practical?
- Is there a theoretical understanding of why directly manipulating parameters can inject backdoors, or is it a byzantine attack?
- How do you calculate the Attack Success Rate (ASR) without mentioning the backdoor task?
- How might the attack perform on other models besides image classifiers?
- Can you discuss potential ways to defend against this attack?
- In 4.2 Attack Settings, why only Model Inversion (MI) and Substitution Dataset (SD)?
- In Table 1, what about the performance of ASR?
- In 4.3, they mentioned the terms unlabeled data and labeled data, while in the overall attack settings (4.2), they considered CIFAR-10 and TinyImageNet (both are labeled datasets). What does it mean unlabeled and labeled here?
- On **FakeBA can evade fine-tuning** (4.3), “the server’s labeled data may significantly deviate from those on the clients’ sides”?
- Results in Figure 1 are on which dataset? CIFAR-10 or TinyImageNet?